# A Comprehensive Review of Patented Technologies to Fabricate Orodispersible Films: Proof of Patent Analysis (2000–2020)

**DOI:** 10.3390/pharmaceutics14040820

**Published:** 2022-04-08

**Authors:** Maram Suresh Gupta, Devegowda Vishakante Gowda, Tegginamath Pramod Kumar, Jessica M. Rosenholm

**Affiliations:** 1Department of Pharmaceutics, JSS College of Pharmacy, JSS Academy of Higher Education and Research (JSSAHER), Sri Shivarathreeshwara Nagar, Mysore 570015, India; dvgowda@jssuni.edu.in (D.V.G.); pramodkumar@jssuni.edu.in (T.P.K.); 2Pharmaceutical Sciences Laboratory, Faculty of Science and Engineering, ÅboAkademi University, 20520 Turku, Finland; jerosenh@abo.fi

**Keywords:** orodispersible films, solvent casting, printing technologies, electrospinning

## Abstract

Orodispersible films (ODFs)are ultra-thin, stamp-sized, rapidly disintegrating, and attractive oral drug delivery dosage forms best suited for the pediatric and geriatric patient populations. They can be fabricated by different techniques, but the most popular, simple, and industrially applicable technique is the solvent casting method (SCM). In addition, they can also be fabricated by extrusion, printing, electrospinning, and by a combination of these technologies (e.g., SCM + printing). The present review is aimed to provide a comprehensive overview of patented technologies of the last two decades to fabricate ODFs. Through this review, we present evidence to adamantly confirm that SCM is the most popular method while electrospinning is the most recent and upcoming method to fabricate ODFs. We also speculate around the more patent-protected technologies especially in the domain of printing (two or three-dimensional), extrusion (ram or hot-melt extrusion), and electrospinning, or a combination of the methods thereof.

## 1. Introduction

Oral drug delivery is still considered the most popular and non-invasive route of administration. About 85% of most-sold drugs in USA and Europe are administered via the oral route [1]. Administration of traditional oral solid dosage forms such as tablets, capsules and syrups, or other liquid formulations, is challenging in pediatric and geriatric patients with dysphagia (functional or psychological), tonsillitis, oral cancer and other diseases [2]. As an alternative to these dosage forms, a relatively novel oral drug delivery system/technology has been developed, denoted ‘oral thin film’ technology [3]. The present review is focused on orodispersible films (ODFs) that do not require mastication and/or water to consume/swallow the medicine. They are very thin, single-layered or multi-layered sheets, portable and attractive dosage forms. When administered to the tongue, saliva hydrates the film to disintegrate rapidly followed by natural swallowing by the subjects for absorption into the blood circulation via the gastrointestinal tract [4]. In general, ODFs can be classified based on the number of layers, release profile, type of active ingredient loaded, and fabrication method employed to prepare ODFs (Figure 1).

Further, ODFs can be used as a carrier for microparticles, nanoparticles or nanocrystals and self-emulsifying systems that control the rate of drug release and thus, also bioavailability [24,25,26]. The most popular and industrially employed method for fabricating ODFsis the solvent-casting method [27,28]. Recently, Musazzi and colleagues carried out a nice review highlighting the trends involved in the production of orodispersible films. They described various methods, namely solvent-casting, electrospinning, hot-melt extrusion and printing methods [29]. Nonetheless, the review by Musazzi and colleagues is silent about patented technologies to fabricate ODFs. Prior to the review by Musazzi, Anna Filipa and colleagues, in 2015, carried out a review on oral thin films and highlighted its intellectual property aspects (patents). Again this review was very broad, focusing on all types of oral films (orodispersible, sublingual and buccal films) [27]. Similarly, Gijare and colleagues carried out a review focused on discussing only the polymer, polymer blends used in the film formulation. Admittedly, even this review was silent on platform technologies per se to fabricate ODFs [30]. Recently, our group has also provided Indian patent perspectives on ODFs, wherein we have focused on highlighting actively researching Indian companies and patent applications filed/granted in India. In addition, we have also highlighted international (under the PCT (Patent Cooperation Treaty)) patenting of inventions by Indian companies [31]. Consequently, we believe that a systematic patent literature search on ODF technologies as a basis for a review is much needed for both industry and academia. The present review is thus novel and first of its kind in the domain of ODFs, which helps one to understand the evolution of different platform technologies to fabricate ODFs. Accordingly, this review is focused on delving into the technological prospection by relying on global patent literature over the last two decades (2000 to 2020).

## 2. Search Strategy and Selection Criteria

Patent literature was searched using the freely available online patent database LENS covering the literature from 1st January 2000 to 31st December 2020 (Figure 2). The search fields employed for the study included title/abstract/claims. The taxonomy of key words employed in the search with different levels and the number of hits included in the present review are tabulated under Table 1.

Patent classification helps the patent office to systematically classify the patent applications based on the technical field to which it belongs. There exist many different classification codes, but the one that is popular and implemented by the European Patent Office and U.S. Patent and Trademark Office is the International Patent Classification (IPC) and the extension of IPC is the Cooperative Patent Classification (CPC), which has 8 sections (A to H), of which each section corresponds to a particular technical field of the patent application. By example, ‘Section-A’ deals with ‘Human Necessities’, under this section, class A61 deals with preparations for medical, dental or toilet purposes. Particularly, A61K9/0056 deals with mouth soluble or dispersible forms and A61K9/7007 deals with drug containing films, membranes or sheets.

The records identified (*n* = 132) were critically analyzed for their relevance and the duplicates were removed (*n* = 38) from the analysis. For instance, citations that dealt with fast-dissolving tablets or oral disintegrating tablets, dental whitening strips were omitted from the study. Hits or the results that were not in English were subjected to machine translation to confirm their relevancy and such citations were accordingly included or excluded from the analysis. In addition, hits wherein ODFs are merely mentioned for the sake of comparison with other formulation types namely orodispersible tablets and/or transmucosal films were eliminated from the study. Further, citations that dealt with chewing gums and hard candies were eliminated as they merely mentioned ODFs as alternative formulations/embodiments and did not explicitly deal with ODFs per se. Citations that dealt with thin film technology were also included in the study. Overall, the search resulted in identifying a total of 132 records/hits that included both published pending patent applications and granted patents. Our attempt to screen the technologies to fabricate ODFs resulted in a total of 33 hits, which were included in the present review. Nonetheless, where relevant, in the present study, a few patent citations that were filed and/or granted before the year 2000 were also included to identify and review certain critical aspects detailed under the following sections of the manuscript.

## 3. History

The early conception of oral thin films is dated back to 1955, wherein a ‘*dope*’—thin film of polymers and plasticizers—was prepared for treating plants and soil [32]. Later in 1962, paper film impregnated with drug solution was reported and patented in Belgium [33]. It was actually in the year 1964 that oral films were first conceived and reduced to practice by Dr. L. Leslie of Ashe Chemicals, UK. He developed oral films, wherein a rice paper was sprayed with hyoscine hydrobromide or a gelatine sheet coated with cyclohexylamine lauryl sulphate. Dr. Leslie went on to propose sandwiched films, wherein gelatine sheets were sandwiched together to hold the drug [34]. This was followed by developments in 1977, wherein Schering Aktiengesellschaft (currently Bayer) developed oral films loaded with dispersed drug for transmucosal drug delivery via the oral cavity, vagina and nasal routes. The drugs that were dispersed in the films were selected from a group comprising sulphonamides, estrogens alone or in combination, gestagens, anti-diabetics, antibiotics and anti-inflammatory agents [35]. Thereafter, the film formulations just remained as a concept until Pfizer launched “*Listerine^®^PocketPaks*”, for breath freshening in the year 2001 [36] and various other products (Figure 3) were launched in the past two decades [37,38]. We now turn towards explaining various patented technologies to fabricate ODFs.

## 4. Patented Technologies

A ‘patent’ is a type of intellectual property right (legal right) that is territorial in nature and has tenure of 20 years, post which it will be open for public to practice the patented invention. Every country has its own patent laws to examine and grant patent rights. A patent per se is a techno legal document that provides complete information about the invention to enable a person skilled in the art to perform the invention post-expiry of the patent right. The key section of the patent specification is ‘claims’ where the crux of the invention to be patented is claimed to seek legal protection. We now examine different patented technologies to fabricate ODFs.

### 4.1. Solvent-Casting Method (SCM)

SCM is the most prominent method employed in the fabrication of ODFs. Generally speaking this method involves the steps (Figure 4) of preparing casting solution of a polymer that has the active pharmaceutical ingredient dispersed in it. The casting solution is de-gassed to remove air bubbles, if any, and casted on a plain, clean and smooth surface (substrate or a glass slab/plain film roll). This is followed by drying at a suitable temperature to facilitate solvent evaporation, peeling and cutting to obtain films of desired size and packaging [39].

SCM is the most popular yet simple and industrially acceptable method. Table 2 provides a list of patented SCMs and combination methods to fabricate ODFs. Though this method is simple it has some limitations that include long drying time, dossing issue and different storage conditions of the final product [40]. Furthermore, this method was leading to ODFs with inaccuracies (Figure 5) such as film shrinkage after drying, brittleness, inaccurate drug dose and lack of uniformity in drug distribution [41], leading to uneven ‘drug islands’ in the film or non-dryable cluster of substances [42]. The challenge associated with long drying (at room temperature) time has led to usage of hot air ovens. Due to this, the films face stress and lead to the problem of the ‘rippling effect’, leading to clumping and lack of uniformity/homogenity of the drug. In addition, drying in an oven may also lead to degradation of drug substances that are thermally sensitive. Further, some films tend to catch moisture, thereby leading to problems for storage and stability [43]. Furthermore, higher drying temperatures also lead to bubble formation in the film and the same is probably due to a phenomena called ‘nucleate boiling’ [44]. Yet another limitation during the fabrication process is mingling or obtaining sticky films [7].

Various patented pathways under the umbrella of SCM developed by different companies with a prime focus on obtaining a stable ODF are comprehensively discussed in the following sections. These individually patented technologies also address one or more inaccuracies detailed above and practically faced during the fabrication of ODFs.

#### 4.1.1. UNM Rainforest Technology

Very recently, Jason Thomas of UNM Rainforest Innovations published a US patent on a relatively novel SCM to fabricate ODFs. In the disclosed method, the proposed film composition comprises athermoresponsive polymer, the drug substance and other excipients that are cast at a temperature at or above the thermal gelation point of the mixture to obtain a stable gel/thin film (Figure 6). Nonetheless, employing higher temperatures in the gelation process leads to formation of air bubbles in the final product (Figure 6a) (especially in systems where degassing was omitted), thereby leading to a product with inconsistent texture. Furthermore, heat sensitive drug substances cannot be employed. In order to overcome this limitation, the inventors proposed employing gelling aids that help reduce the thermal gelation temperature for casting heat sensitive drugs (Figure 6b). For instance, the inventors employed compounds that contain ‘salt-out cations’ namely barium, calcium, magnesium, etc., and ‘salt-out anions’, namely aspartate, 2-carboxybenzoate, dimethylphosphate, etc., either alone or in combination with one another. In this patent disclosure, the term ‘thin film’ is used in contrast to the term ‘gel’. The solvent in the gel is removed by evaporation so as to obtain a dry film [45].

It is pertinent to state here that Acevedo and colleagues studied the gelation mechanism and the interaction of hydroxypropyl methyl cellulose (HPMC) polymer with surfactants and drugs (griseofulvin), which is essential for designing and optimizing a pharmaceutical formulation with the desired performance [46]. Along similar lines, the inventor Jason Thomas employed HPMC as a polymer at a concentration of about 6%, which had a thermal gelation temperature (TGel) of about 32 °C when determined using a rheometer. Similarly, when the polymer was combined with drugs such as ibuprofen or itraconazole, the TGel of respective combinations was 36 °C and 47.5 °C. Increase in dose of the drug was also leading to increase in TGel value. For instance, the thermal gelation temperature of HPMC and ibuprofen (120 mg) had aTGel of 36 °C, which increased to 38 °C when the dose was increased to 400 mg. Furthermore, the TGel of naloxone when combined with 2% HPMC was about 69.1 °C. The fabricated ODFs of ibuprofen and itraconazole had an average disintegration time of less than 30 s. Most importantly, besides the addition of gelling aid, ethyl cellulose (0.25% *w*/*v*) was also added to the casting solution before gelation of the drug and polymeric mixture to basically facilitate the incorporation of hydrophobic ingredient (drug). Optionally, the drug (for instance, naloxone) was also sprayed on the final fabricated film to facilitate initial immediate release followed by slow release. As regards the film properties, it has an average disintegration time of less than 30 s. The average tensile strength of the films was 24.4 N/mm^2^.

#### 4.1.2. VersaFilm Technology

One of the prime reasons for lack of homogeneity (or non-uniformity) of drug distribution in the film formulation is primarily due to aggregation or agglomeration and recrystallization phenomena seen with the excipients or the active substances present in the film formulation. For instance, Bruce Caroline, inventor of the Novartis patent on nicotine thin films hypothesized that an increase in moisture content leads to enhancement in the crystallization of polyethylene oxide, which leads to an increase in disintegration time of the film when compared to a non-crystalline amorphous film [47]. In order to overcome such a challenge, IntelGenx has proposed employing ‘liquid crystallization inhibitors’. It refers to any substance that exists as a liquid at 2.7 °C+ crystallization inhibitors that has the potential to inhibit the crystallization of the active substance in combination with other agents [48]. In order to experience a pleasant feeling when ODFs are administered to the tongue, IntelGenx developed instantly wettable and rapidly disintegrating ODFs by SCM. In the proposed method, it employs a polymer combination comprising hydroxypropyl cellulose and polyvinylpyrrolidone along with triacetin as a plasticizer. In addition, titanium dioxide was used to enhance the rate of disintegration of the ODF. The ratio of polymer combination to titanium dioxide ranges from 3:1 to 5:1. The ODFs of rizatriptan, alprazolam and palonosetron were disclosed. The SCM method employed is quite conventional, wherein the casting solution is casted over a non-siliconized polyethylene terephthalate film, non-siliconized kraft paper or polyethylene film. Most importantly, the method employed eliminates usage of surfactant and polyalcohol [49].

#### 4.1.3. PharmFilm Technology

Aquestive Therapeutics developed an altogether new platform technology called ‘PharmFilm’ to fabricate all types of oral thin films including ODFs [40]. The patented apparatus (Figure 7) comprises a feed tank for loading and mixing the composition of the film formulation. The first set of metering pump and control valves helps in controlling the release of the film formulation. The active pharmaceutical ingredient along with or without flavoring agent is introduced via the mixer. The second set of metering pump controls the release of the final mixed formulation comprising both the API and excipients. They are released into the pan and the final ODFs of desired thickness are obtained with the help of metering and support rollers on a substrate. The film is dried (Figure 7) from the bottom side of the substrate to obtain films with less drug non-uniformity and moisture.

#### 4.1.4. Melting Film Technology

Hexal has created a cluster of six patents that dealt with fabrication of ODFs. For instance, its first patent dealt with single layered and cavity free (region filled with air bubbles and/or liquid(s)) ODFs of anti-emetics and anti-migraine drugs. Nonetheless, the entire patent specification disclosed a single example of fabricating the anti-migraine drug naratriptan using a polymer mixture comprising hydroxypropyl methylcellulose, ethyl cellulose, 1,3-butanediol, isopropyl palmitate, and paraffin oil as film forming agents. D-sorbitol was used as a sugar and ethanol/water was used as a solvent. The procedure involved the first and foremost step of dissolving D-sorbitol in water followed by addition of 1,3-butanediol, isopropyl palmitate, paraffin oil and ethanol and was stirred continuously. Thereafter, hydroxypropyl methylcellulose, ethyl cellulose and naratriptan were added under continuous stirring. Finally, the mixture was casted over a polyethylene foil and dried at a temperature of 50 °C to remove ethanol/water and the dried film is cut into desired shape and packed [50]. The same process was also employed by Hexal in fabricating ODFs of a neuroleptic drug olanzapine. Here, a comparison of the stability of olanzapine film formulation with its tablet coated formulation was carried out and the study results confirmed that film formulation was associated with less (0.03%) impurities when compared with impurities found (0.73%) in the tablet formulation [51].

Hexal disclosed a method to fabricate ODFs of aripirazole, a drug that tends to crystallize (forms new solvates with low solubility) when dissolved in certain solvents, thereby affecting its release (bioavailability) when administered orally. Therefore, there is a need to develop new polymorphs or solvates of aripirazole that have the potential to not only withstand the process used to prepare its formulations but also stable enough to release from the finished formulation. In this patent, Hexal formulated oral film of aripiprazole anhydrate (Form X) was developed and compared with films formulated using aripirazole monohydrate. Most importantly, the casting solution to fabricate ODFs of aripiprazole anhydrate (Form X) was prepared using a polymer combination comprising microcrystalline cellulose, methylcellulose and hypromellose, and glycerol and tween 80 as plasticizers. The polymer solution with drug was casted and dried at a temperature of 50 °C for 45 min. On the contrary, the ODFs of aripirazole monohydrate were fabricated by using a different conventional casting solution. Both the fabricated ODFs, when tested for their release profile, and the films of aripirazole anhydrate were found to have superior drug release rate (more bioavailable) when compared with the monohydrate version of the film [52].

Hexal went on to propose a new SCM that has the potential to load higher drug dose (70% by weight) and still achieve films with excellent content uniformity or films with no drug island. The model drug used in the process was sumatriptan succinate, an anti-migraine drug. It employs a combination of two organic solvents, namely acetone and ethanol. The method comprises steps of dissolving grapefruit flavor and levomenthol in acetone, followed by addition of sucralose and propylene glycol under continuous stirring. Thereafter, the drug substance, sumatriptan succinate, and the polymer hydroxypropyl cellulose were added and the entire mass was allowed to swell for about 12 h. The swollen mass was coated on a suitable coating plant and dried at a temperature of 40 °C. The dried films had acceptable residual solvent level of about 2000 ppm, which is pharmaceutically acceptable [53].

Hexal also proposed yet another robust and economical SCM that has the potential to fabricate ODFs that are stable against forming impurities over an extended period of storage time. This patent disclosed ODFs of combination drugs namely buprenorphine hydrochloride and naloxone at a ratio ranging from 4:1 to 3.5:1. Polyethylene oxide and hypromellose were employed as film-forming agents and maltitol and acesulfame K were sweetening agents. Tri-sodium citrate and citric acid were buffers. Lemon oil was used as a flavorant and dye (FD&C yellow No. 6) was used as a coloring agent. The steps involved in the process included preparing a suspension of buprenorphine and naloxone followed by adding the polymer. The organoleptic agents (flavor, sweetener and colorant) were added. A film was casted using this mixture and was dried at a temperature of 65 °C for 15 min. The dried film was cut into desired size and shape. A similar film formulation was also prepared by replacing lemon oil with artificial lemon flavor. The fabricated films were subjected for stability testing and were analyzed by HPLC method for presence of impurities. The study results confirmed that the films with natural lemon oil showed a lower percentage of impurities (<0.71%) while the other composition with artificial lemon flavor showed a higher percentage of impurities (6.12%) [54]. As an extension of this invention and in order to enhance the bioavailability of buprenorphine hydrochloride, Hexal filed another patent application. In its new patent application it employed buprenorphine of different particle diameters: DV90 = 90 micrometers, DV50 = 30 micrometers, DV10 = 6 micrometers, while the rest of the ingredients of the composition remained the same as disclosed in its previous patent on films of buprenorphine hydrochloride and naloxone. The processing steps remained conventional, such as preparing drug solution, polymer solution and combining both under continuous stirring followed by addition of all the remaining excipients. In this patent, Hexal proposed a novel drying technology that employs three hot air dryers. The first and second dryer each had a length of 1.5 m and the temperature was set to 45 °C and 55 °C, and the time set in both the dryers was 3 min. Similarly, dryer 3 had a length of 2.5 m with a temperature of 65 °C for 5 min. Nonetheless, the drying of ODFs was carried out either in a batch or semi-continuous or continuously using different drying temperatures as proposed above, and most preferably they were dried at a temperature ranging between 40 °Cand 70 °C for a time period between 10 and 25 min. The dried film contained residual solvent, acetone, of about 5000 ppm and the water content was between 3 to 6 weight % [55].

Overall, drying of the casted film is the key to obtain good quality ODF. The patent proposed a novel and inventive drying technology with specific temperatures to overcome various inaccuracies associated with films fabricated by SCM. In addition, it also provided novel SCM methodologies that help prevent formation of ‘drug islands’ or non-dryable cluster of substances. In short, the proposed method could obtain ODFs with good uniformity in drug content. Hexal was also successful in proposing an improved method to fabricate ODFs of drugs (aripiprazole) that tend to crystallize in some solvents.

#### 4.1.5. Perforated Film Technology

In yet another interesting US patent, inventor Ramesh Bangalore, disclosed rapidly dispersing/dissolving perforated ODFs that have the potential to load higher drug dose using polymers and plasticizers [6]. The prime reason behind creating perforations in the ODFs was to achieve rapid disintegration even when the thickness of the film is increased to load higher drug dose. The process involved in the preparation of perforated ODFs (Figure 8) begins by first preparing the formulation composition comprising polymers, plasticizers, organoleptic agents and various other pharmaceutically acceptable excipients and the drug substance. Once the formulation composition is prepared, it is loaded onto the steam jacketed blender/mixer and is kept ready for loading onto the polymer film roll, which acts as a backing matrix. The loaded formulation composition is spread using a doctor knife (spreader) for even distribution of predetermined thickness. This stage is followed by drying with the help of a hot air oven for drying the film. Alternately, the drying step is also accomplished using laser energy, irradiation or infrared lamps and heating coils. The dried film is rolled up with an additional barrier layer film to prevent the two coated layers from coming in contact with each other (mingling). Thereafter, when needed, perforations on the film are made by a suitable mechanism using a perforator followed by packaging and storing. The thickness of the final film ranged from 50 to 10,000 microns and the disintegration time ranged between 1 and 600 s. As regards the drug dose, it ranged from 10 micrograms to 500 milligrams. The disclosed technology is not just limited to fabricate ODFs, instead it can even be employed to fabricate wound dressing patches, buccal patches, intra-vaginal patches and transdermal patches as well. Lastly, the perforated ODFs can be in any shape ranging from square, oval, zig-zag, round, triangle, twisted, ribbon shaped and oblong with predetermined size.

#### 4.1.6. Rapid Film Technology

Labtec has three international patent applications under the patent cooperation treaty on its technology. The focus in the three patent applications is primarily towards developing a standard casting solution. In its first patent application, it disclosed a polymer combination comprising polyvinyl alcohol and polyethylene glycol to fabricate non-mucoadhesive film dosage forms or ODFs of donepezil, ondansetron and various other drug substances [56]. Following this, they further modified the casting solution to comprise polymers, namely polyvinyl alcohol, polyethylene glycol and rice starch, pH adjusting agent, namely sodium or potassium carbonate or bicarbonate, sodium hydroxide, and potassium hydroxide, and an anti-microbial essential oil, namely peppermint oil, eucalyptus oil, crisp mint oil, spearmint oil, and *Pelargonium sidoides* root extract, either alone or in combinations thereof [57]. As an extension of this work, they disclosed yet another casting solution, this time with the objective of preventing drug degradation in the film formulation by employing a reducing agent. The drug in question was zolmitriptan and the reducing agent employed in the formulation was selected from a group comprising glutathione, methionine, thioglycerol, thio lactic acid, methionine and dithiothreitol. In the overall study, glutathione was found to be ideal at a concentration of 0.2 to 1.0 weight % [58]. Furthermore, Labtec in collaboration with APR Applied Pharma research carried out bioequivalence studies between the ODFs fabricated by Rapidfilm technology and traditional orodispersible tablet formulations. The results of the study confirmed that ODFs offer higher comfort and convenience for consumption by the subjects [59].

#### 4.1.7. Thinsol Technology

Here again the focus was on developing a standard casting solution comprising enzymatically digested carboxy methyl cellulose, emulsifiers such as lecithin or polysorbate 80, plasticizers selected from a group comprising sorbitol, propylene glycol or glycerol, thickeners such as modified corn starch, and maltodextrin, either alone or in combination with one another. Humectant such as propylene glycol was also employed in preparing the casting solution. Furthermore, alkalizing agent such as sodium bicarbonate was also employed, along with organoleptic agents. The model drug substances describedin the patent application were caffeine, nitro-glycerine, bio ferrin and vitamin D3 [60].

#### 4.1.8. Cure Film Technology

Cure proposed a new polymer combination to fabricate ODFs that helped achieve rapidly dissolving films (<60 s) with moisture content of less than 8%. The film composition comprises binders–polyvinyl alcohol, polyvinyl alcohol–polyethylene glycol copolymer and film-forming polymer hydroxypropyl methyl cellulose. Optionally, the composition also contains moisture deterring agent Eudragit and pH adjusting agent citric acid [61].

#### 4.1.9. Smart Film Technology

Generally speaking, enhancing the dose of the drug in ODF often leads to a bitter taste in the mouth, thus leading to immediate spitting by the subjects. In order to overcome such a disadvantage, Seoul Pharma developed a technology wherein at least 100 mg of sildenafil citrate was successfully loaded onto an ODF. The taste masking of sildenafil was achieved by using a mixture of magnesium oxide and sodium hydroxide in a ratio of 1:4 to 4:1 (preferably 1:1). Furthermore, they also employed pullulan as a film forming agent and a combination of propylene glycol and polysorbate as a plasticizer [62].

#### 4.1.10. Other Technologies

Pharmathen S.A., Greece has disclosed stable orodispersible formulations of enalapril maleate for treatment of hypertension in children [63]. The process employed is SCM, wherein a combination of polymers namely pullulan and modified starch are used in a ratio ranging from 1:1 to 1:2. Glycerol (5 to 10% *w*/*w*) was used as a plasticizer along with a surfactant polysorbate 80 (4 to 6% *w*/*w*) and sucralose as a sweetener (4 to 7% *w*/*w*). It also employs the pH increasing agent sodium hydroxide at a concentration of about 1 to 3% *w*/*w*. The surprising technical effect seen here is the stability of enalapril, wherein its stability is significantly enhanced by employing a simple alkaline agent (sodium hydroxide). Furthermore, an increase in pH also helped the formulated ODF to improve its relative humidity and the disintegration time. The pH of the final ODF ranged from 6.3 to 6.7 (preferably 6.4 to 6.5) and had a disintegration time of 19 s. As regards the processing steps to fabricate ODFs of enalapril maleate, the polymer combination, pH enhancing agent, surfactant, plasticizer, sweetener and water were mixed under a nitrogen atmosphere until a homogeneous casting solution was obtained. The obtained film was casted on a polyethylene silicon coated paper and dried at a temperature of 40 °C for about 4 h. Thereafter, the dried film was cut into single dose films of suitable size and packaged in aluminum foils. The ODFs of enalapril were produced in various dose strengths of 1.25 mg, 2.5 mg, 5 mg, 10 mg and 25 mg of enalapril for subjects between 1 to 18 years of age. Lastly, SCM was also used to fabricate ODFs of herbal active agent, namely garlic (comprising allicin) [64] and a pine mushroom, *Sparassiscrispa*, extracts [65].

#### 4.1.11. SCM + Printing Method—Structured Orodispersible Films (SOFTs)

This technology was developed by LTS Lohman of Germany, and was published as non-patent literature in 2019 [66] and as an unexamined USPTO patent application in 2020 [67]. SOFTs basically refer to drug free templates fabricated by SCM. The drug of interest, which is either in solution form or suspension form, can be printed on the upper porous surface, while the lower end is closed to prevent leakage of the printed drug substance (Figure 9). SOFTs help in higher drug loading and are prepared using water-soluble cellulose derivatives. Post printing the drug of choice, a protection layer is applied to enhance the handling/safety, which inhibits direct contact of the drug substance. Overall, this technology is a combination of SCM and printing methods for high drug loadings. In yet another disclosure, ZIM Laboratories Limited, India, has filed an international patent application under the PCT claiming drug printing by ink-jet methods on oral films fabricated by SCM [68]. The key aspect of the printing process was the ink formulation, whose viscosity (8 to 12 m Pa s) and surface tension (1–1.5 N/m) are vital. Different drugs that were printed include diclofenac sodium, clonazepam, levocetrizine, nifedipine and loperamide. The fabricated films had a disintegration time of less than <14 s.

**Table 2 pharmaceutics-14-00820-t002:** Patents on solvent casting method and its combination methods.

Sl. No.	Country Code	Publication/Patent Number	Title	Applicant/Assignee	Brand Name	Ref.
1	US	US20090047350	Perforated water soluble polymer based edible films	Bangalore Ramesh		[6]
2	US	US7425292	Thin film with non-self-aggregating uniform heterogeneity and drug delivery systems made from there from	Aquestive Therapeutics Inc	PharmFilm	[40]
3	US	US20210022990	Thermally gelling drug formulations	UNM Rainforest Innovations	-	[45]
4	US	US20110136815	Solid oral film dosage forms and methods for making same	IntelGenx	Versa Film	[48]
5	US	US9301948B2	Instantly wettable oral film dosage form without surfactant or polyalcohol	IntelGenx	Versa Film	[49]
6	US	2008/0213343	Oral, Quickly Disintegrating Film, which Cannot be Spit Out, for an Antiemetic or Antimigraine Agent	Hexal Pharmaceuticals	Melting Film	[50]
7	US	2008/0200452	Oral, Rapidly Disintegrating Film, Which Cannot be Spat Out, for a Neuroleptic	Hexal Pharmaceuticals	Melting Film	[51]
8	US	2012/0149713	Oral films comprising 7-[4-[4-(2,3-dichlorophenyl) piperazin-1-l]butoxy]-3,4-dihydro-1h-quinolin-2-one base or salts or hydrates thereof	Hexal Pharmaceuticals	Melting Film	[52]
9	EP	EP2632443B1	Preparation of orodispersible films	Hexal Pharmaceuticals	Melting Film	[53]
10	WO	WO2014/076117	Orodispersible film compositions	Hexal Pharmaceuticals	Melting Film	[54]
11	EP	EP2886103 A1	Pharmaceutical orodispersible film comprising buprenorphine particles with a particular size	Hexal Pharmaceuticals	Melting Film	[55]
12	WO	WO2008040534	Non-mucoadhesive film dosage forms	LabtecGmbh	Rapidfilm	[56]
13	WO	WO2009043588	pH regulating antibacterial films for the oral or vaginal cavity	LabtecGmbh	Rapidfilm	[57]
14	WO	WO2011124570	Oral film formulations	LabtecGmbh	Rapidfilm	[58]
15	WO	WO2009055923	Ingestible film composition	BioEnvelop	Thinsol	[60]
16	WO	WO2020014431	Rapidly disintegrating oral film matrix	Cure Pharmaceutical	Cure Film	[61]
17	US	US10092651	High-content fast dissolving film with masking of bitter taste comprising sildenafil as active ingredient	Seoul Pharma Co Ltd.	Smart Film	[62]
18	US	US20200360461	Orodispersible film composition comprising enalapril for the treatment of hypertension in a pediatric population	Pharmathen SA	-	[63]
19	IN	2199/MUM/2015	Stabilized orodispersible film and its preparation	Dnyaneshwar R Pawar	-	[64]
20	KR	KR20200069611	Orodispersible films using fermented extract of *SparassisCrispa*	Agricultural Company Corp Smart F&B Co. Ltd.	-	[65]
21	US	US20200108011	Structured orodispersible films [SOFTs]	LTS Lohmann Therapy Systems	-	[67]
22	WO	WO2019198105	Composition of active ingredient loaded edible ink and methods of making suitable substrates for active ingredient printing on orodispersible films	Zim laboratories Ltd.	Thinoral	[68]

Note: **IN:** India; **KR:** Korea; **CN**: China; **US:** United States; **EP:** Europe; **WO:** International patent application under the Patent Co-operation Treaty (PCT).

### 4.2. Hot-Melt Extrusion (HME) Method

HME is all about melting and mechanically processing polymeric materials above their glass transition temperature (Tg) to effect molecular mixing of polymers, drug substances and other agents [69]. The machinery employed in HME is called ‘extruder’ and is used primarily in polymer processing industries [70]. Nonetheless, it is also used in fabrication of various pharmaceutical formulations [71]. The term extrusion or extrude refers to pushing or forcing it out. In this process, the polymer or its composition is melted and forced via the die (orifice) of a particular cross-section and cooled or subjected to downstream ancillary equipment for further processing [72]. Different types of extruders exist (Figure 10A) and the common types include ram-extruder, single-screw and double-screw extruders. The double-screw extruders are further classified based on its design (parallel or conical) and rotation (co-rotation or counter-rotation), either with, without or close intermeshing (Figure 10B) [72]. The extrusion process is carried out using an extruder witha temperature controlled barrel and uses either a single-screw or twin-screws. The screws rotate and help in not only conveying the material via the barrel but also help in melting and mixing the blends feed from the feeder of the extruder. For instance, a single screw helps in melting and conveying while a twin screw when co-rotating helps in mixing and conveying through the barrel [73]. A modular screw has different functions and zones (Figure 10C) that help to perform different functions ranging from particle-size reduction, mixing and conveying functions [74]. In short, it helps in extruding the feed material. The ODF composition (polymer blend with actives), when fed (flood or starve feeding) via the hopper, falls onto the rotating screw(s) from where it is conveyed to the mixing/kneading zone to undergo melting, which is further mixed in the final zone. The pressure generated in the extruder forces the extrudate out from the extruder via the orifice of the die [75].

HME is a continuous, single step and solvent-free ODF fabrication process. It ensures both distributive and dispersive mixing and thereby guarantees excellent drug content uniformity [76]. It is also a potential alternative to SCM as it helps in overcoming some of the limitations such as eliminating usage of solvents, mixing and drying steps [77]. As a technology (Figure 10D) it processes the polymer blends and crystalline drug substances using shear forces and temperature to obtain amorphous extrudate via the die of a desired design [78,79]. HME is an efficient and cost effective process. Another advantage of HME is that it helps in turning the poorly soluble crystalline drug substances (poor absorption and low bioavailability) to an amorphous solid dispersion or liquid solution with good absorption and bioavailability [80,81]. While it has various advantages, it also has some common limitations (Figure 11) encountered during the fabrication process [73]. The limitations include, but are not limited to, die swell phenomenon, fish eye effect seen during blending, bubbles in the extrudate and bambooing effect, wherein the extrudate surface is rough like shark skin. Yet another limitation of this technology is its high process temperatures as it could directly impact polymers/drugs employed in the formulation [40]. However, such a limitation was successfully handled by Pimarande and team by employing a single-screw based HME (110 °C) to obtain ODFs of chlorpheniramine maleate with a disintegration time of 6 to 11 s [77]. In the following, we turn towards explaining patents that deal with fabrication of ODFs by HME method.

#### Patented HME Technologies

Recently, BonAyu life sciences disclosed the twin-screw HME process to fabricate ODFs of various active agents. Polymer blends comprising maltodextrin and hydroxypropyl cellulose were employed in a ratio ranging from 1:3 to 3:1. Organoleptic agents (coloring, flavoring and sweetening) were also added to this saliva stimulating agent to obtain a mixture. Thereafter, plasticizer and anti-sticking agents were also added and mixed to obtain a final mixture, which was extruded using twin-screw HME at a temperature ranging between 80 to 110 °C, preferably between 85 to 95 °C. Similarly, the twin-screw rotations were maintained in the range between 30 rpm to 90 rpm to increase the residence time and shearing stress of the mixture in the barrel to obtain a uniform and smooth mixture from the conveying zone to the die zone. Nonetheless, the method disclosed did not provide a crystal clear picture on obtaining ODFs after obtaining the extrudate [82]. Similarly, Chongqing Runze Pharmaceuticals disclosed a HME method to fabricate ODFs of levo-oxiracetam. The HME processing temperature ranged from 85 °C to 90 °C.The disintegration time of the final ODF was less than 32 s [83].

Novartis disclosed a novel method to fabricate drug coated granules as ODFs by hot-melt extrusion technique [84]. The key feature of the method disclosed is that it employs mild processing conditions such as low temperature, pressure, and shear by adjusting the extruder screw speeds. The prime reason behind employing such mild processing conditions is to prevent the degradation of the coating on the drug substance. Maintaining the drug coating helps preserve the properties of the drug in question and it even prevents its leaching from the coating material due to shear stress of the extruder screw. For instance, the properties could include, but are not limited to, taste masking, controlled release and maintenance of drug loading capacity that rapidly dissolves in the mouth and enters the stomach naturally along with the saliva.

The composition disclosed (Table 3) comprises a polymer, polyethylene oxide, with a molecular weight ranging from 70,000 to 230,000 Daltons, 5 to 35% sugar alcohol (sorbitol or mannitol or a combination of both) with a melting point of more than 75 °C, 5 to 20% by weight of polyethylene glycol with a molecular weight ranging from 100 to 4000 Daltons, as plasticizer, and 10 to 75% of coated drug substance in granular form. It also comprises various organoleptic agents in desired concentration ranges. While it disclosed ODFs of various drug substance, the one that is extensively tested was dextromethorphan hydrobromide coated with ethyl cellulose or cellulose acetate. The coated granule size ranged from 80 to 200 microns.

Depending on the ingredients of the composition, the processing temperature is decided. For instance, the melt temperature employed while fabricating polymer coated dextromethorphan hydrobromide ranged from 50 °C to 70 °C. The thickness that final ODFs obtained ranged between 0.05 millimetres to 2.00 millimetres (preferably 0.1 to 0.8 millimetres thick). The above composition was subjected to hot-melt extrusion by employing suitable process parameters (Table 3) to obtain a thin sheet followed by cutting to obtain thin strips. The entire process was maintained at mild conditions and was never more than the melting point of the sugar alcohol.

Inventor Joseph Fuisz disclosed orally dissolving sustained release thin films, buccal and sublingual films of nicotine (snuff/tobacco), which were fabricated by the HME method. Hydroxypropyl cellulose was employed as a polymer at a concentration of >20 weight % of the total composition and tobacco. In one of the embodiments of its disclosure, it used a single screw extruder, and the temperature was set at 110 °C for the initial zone and 149 °C for the following subsequent zones and the slot die. The screw was set to a speed of 180 rpm. The dry blend was fed at a rate of 7kg per hour and the liquid flavor was vented from the extruder. Overall, the residence time of the blend in the extruder was 90 s. The width of the slot die was set to ten inches and the film obtained had a thickness of 13 mils, which was rolled onto a roller. The sheets obtained were flexible and were cut into desired dimensions [85]. The inventor Joseph Fuisz also disclosed methods to scale-up fabrication of nicotine films by HME method (Figure 12).

Apart from the above, our comprehensive patent search and review did not result in any ODFs that were fabricated by patented HME process. Nonetheless, we found that Novartis patented fabrication of buccal thin films containing nicotine, wherein said films release nicotine over a sustained period of time [86]. This patented technology of Novartis could probably even be applied to ODFs as well. While HME offers various advantages over other methods of fabrication, it does have some limitations. Therefore, one has to judiciously select and employ HME method depending on various characteristics of the drug substance and polymers.

### 4.3. Electrospinning Method

Electrospinning is a relatively novel method employed in fabrication of ODFs. This method has a basic setup (Figure 13) and the components include a syringe pump, high-voltage power supply (direct or alternating current), a spinneret—nothing but a hypodermic needle having a blunt tip—and a collector to collect the nanofibers. During the electrospinning process, the polymer solution is extruded via the spinneret to produce a droplet and upon applying electric field leads to deformation of the droplet into a Taylor cone, from which a charged jet is ejected and stretched into thin solid nanofibers, which are collected onto a rotating drum [87,88].

In 1902, inventors John Cooley and William Morton were the first to file patents describing a prototype of the setup for the electrospinning method [89,90]. Following this, the inventor ‘Formhals Anton’ also filed two US patents pertinent to the developments/improvements in this method of electrospinning method to fabricate textile yarns [91,92]. While this method per se has pretty diversified applications, its applications in the domain of drug delivery have been very promising so far. For instance, this method can be employed in fabricating sustained release nanofibers of tetracycline hydrochloride [93]. This method was also found to be promising and progressive in fabrication of immediate release dosage forms as it tends to improve dissolution and bioavailability of poorly-soluble drugs [94].The critical requirement to employ the electrospinning method for ODFs is the preparation of the final formulation in the form of a solution. The challenges in this process often arise from solubility, stability (biphasic mixture) and taste point of view. One can employ solvents such as ethanol, methanol, dichloromethane, tetrahydrofuran and chloroform. However, they are not considered as ideal due to safety issues.

Zentiva KS, a company from Central Europe, filed two European Patent Applications for preparation of tadalafil ODFs by electrospinning method. In its first patent, it employed a solution formulation [95], while in the second patent application, quite contrary to what is disclosed in the first patent, it employed a suspension formulation of tadalafil to prepare ODFs by electrospinning method [96]. It is pertinent to state that Zentiva KS employed a patented Nanospider^TM^ electrospinning technology (Figure 14A) to fabricate ODFs by electrospinning method [97]. This method is a needle-free, high voltage, free liquid surface electrospinning process, wherein it employs a stationary electrode system. The carriage distributes the spinning solution over the surface of the stationary electrode by reciprocating motion (Figure 14B). Once the desired voltage is applied, the nanofibers start forming and are collected onto an edible substrate as a film that can be cut into desired size. The patented electrospinning processes of Zentiva helped fabricate ODFs by solution and suspension method (Figure 15).

### 4.4. Printing ODFs

Yet another method of fabricating ODFs is by printing technologies—two dimensional (2D) (ink-jet printing), three dimensional (3D) or additive printing and flexographic (or flexo) printing methods. For instance, in ink-jet printing (IJP) (Figure 16), the process begins by preparing the ink formulation that is to be printed over an edible substrate. Depending on the type of printer head, the ink is ejected from the print head onto the surface of the edible substrate. Post printing, the printed orodispersible films (POFs) were dried and cut into desired shape and size followed by packaging and storing for usage by the subjects who are in need thereof.

Our group has carried out a comprehensive review on printing methods in the production of ODFs [98], wherein we discussed various printing methods namely IJP, flexo, 3DP or additive manufacturing methods, namely filament deposition modeling, hot-melt ram extrusion three dimensional printing and semisolid extrusion 3D printing. Further, we have also discussed various limitations associated with printing of drugs as ink formulation onto the surface of the substrate (Figure 17).

Turning now towards patented technologies, Roquette Freres SA disclosed an ink-jet printing method to fabricate ODFs of meloxicam, dextromethorphan hydrobromide and loperamide. Here the ink formulation had a viscosity ranging from 10 to 5500 mPa. Sand was directly printed onto a blister pack that hasa cavity to hold the film followed by drying to remove the solvents and hermetical sealing (Figure 18). The temperature employed during printing ranged between 40 °C to 45 °C. Here plant based polymers such as pea starch were employed as a polymer forming material [99].

TESA Labtec protected its invention with USPTO on flexographic printing of ODFs both in laboratory and industrial settings. Flexo is an offset, rotary printing process (Figure 19) to fabricate POFs. Broadly, this process has two important steps: (a) fabricating drug free ODF layer; and (b) drug printing on the drug free ODF layer by the flexo process.

(a)**Drug free ODF layer:** The drug free ODF layer is prepared by solvent casting method using hydroxypropyl methylcellulose and polyvinylpyrrolidone as polymers. Glycerol was used as a plasticizer and water as a solvent. The casting solution of polymer was casted on an intermediate liner with the help of a coating machine followed by drying in an oven with four heating-zones. The dried drug free film was rolled up to form a jumbo roll, which was later cut into daughter rolls with a width of 2 cm and length of about 100 m.(b)**Flexo printing:** The first step in this process is preparing ink formulation to be printed on drug free ODF layer. The ingredients employed in the ink formulation were hydroxypropylcellulose, brilliant blue and ethanol. Drugs such as rasagiline mesylate and tadalafil were employed as model drugs. Rasagiline was soluble in the ink formulation while tadalafil was formed as ink-suspension. The flexography printing machine is equipped with anilox roller (capacity of 11.71 cm^3^/m^2^ or 80 cm^3^/m^2^) cells withdifferent geometries (Figure 19A) that help in metering the ink and the excess amount of ink is carefully removed using the doctor blade system. The ink solution (or suspension) of respective drug substance(s) was movedfrom anilox roller to printing cylinder and then to the impression cylinder to print the ink on drug free ODF layer, which was passed between the printing and impression cylinders. Thereafter, the solvent was removed using a fan and the final ODF was rolled up, and later on they were cut into desired dimensions and shapes. The flexo printing was carried out at a speed of 16 m/min and was repeated four times to obtain the drug printed ODFs. Overall, this method helps in fabricating ODFs of highly potentlow-dose drugs and also drugs that are heat sensitive. It can even help printing drug solution and suspensions and produces a homogenous distribution. Drugs such as rasagiline mesylate and tadalafil were successfully fabricated by the flexo method and the fabricated films had a disintegration time of < 45 s [100,101].The proposed method can be employed in a laboratory setting to fabricate personalized ODFs or it can also be employed in an industrial setting as well.

In addition to the above disclosed flexo printing process, TesaLabtec disclosed and patented yet another flexo process that has the potential to not only produce ODFs continuously but is also capable of monitoring them continuously [102]. Continuous monitoring helps in gaining control over the fabrication process and helps minimize loses. First and foremost, the printing ink solution consisting of hydroxypropylcellulose, the desired active pharmaceutical ingredient, blue dye and ethanol were combined to obtain a solution or a suspension. The flexo printing machine is fitted with a chromed gravure roll (54 lines per cm, well depth of 40 µm and a scooping volume of 11.7 cm^3^/m^2^) and an Ethylene Propylene Diene Monomer coated rubber roll as a printing cylinder. The opposing/impression cylinder combines with the printing cylinder to form a roll nip to print the drug containing ink solution onto a drug free polymeric (hydroxypropyl methylcellulose) ODF layer (Figure 19A). The printing speed was 15 m/min and was printed thrice in succession. The drug printed layer is allowed to enter the drying unit to remove ethanol and the dried film is rolled and cut into desired sizes and characterized.

As regards monitoring, the printing ink solution comprising an admixture of dye and drug substance is determined by physical method of transmission measurement, wherein the dye absorbs the light (red luminescent light of wavelength 650 nm) emitted from the source (Figure 19B). When a small portion of the ink solution (dye and drug admixture) is printed, it absorbs a minimal amount of light and vice versa when a high amount of ink is printed. Therefore, the percentage transmittance depends on the quantity of the dye and the associated drug thereof. In short, the % transmission decreases with the increase in dye and drug admixture. As evident from the results, the blank or drug free ODF has % transmittance of 80.9%, first cycle print—74.7%, second cycle print—64.7% and third cycle print—56.7%. Accordingly, the drug concentration when determined by chemical method increased with each cycle of the print, giving 0.34 mg/6 cm^2^, 0.63 mg/6 cm^2^ and 0.87 mg/6 cm^2^ respectively. Deviations in percentage transmission values beyond the specified limits help in quick identification and intervention to fix the issues in the fabrication process. Recently, 3D printing of low moisture and rapid disintegration ODFs of desmopressin were disclosed in some embodiments of a US patent publication of Cure Pharmaceutical, USA. The printed ODFs disintegration time ranged between 15 s and 20 s with moisture between 4 to 6 wt % [103]. Nonetheless, the disclosed aspects are not comprehensive enough to enable a person withordinary skill in the art to arrive at 3D printed ODFs.

**Figure 19 pharmaceutics-14-00820-f019:**
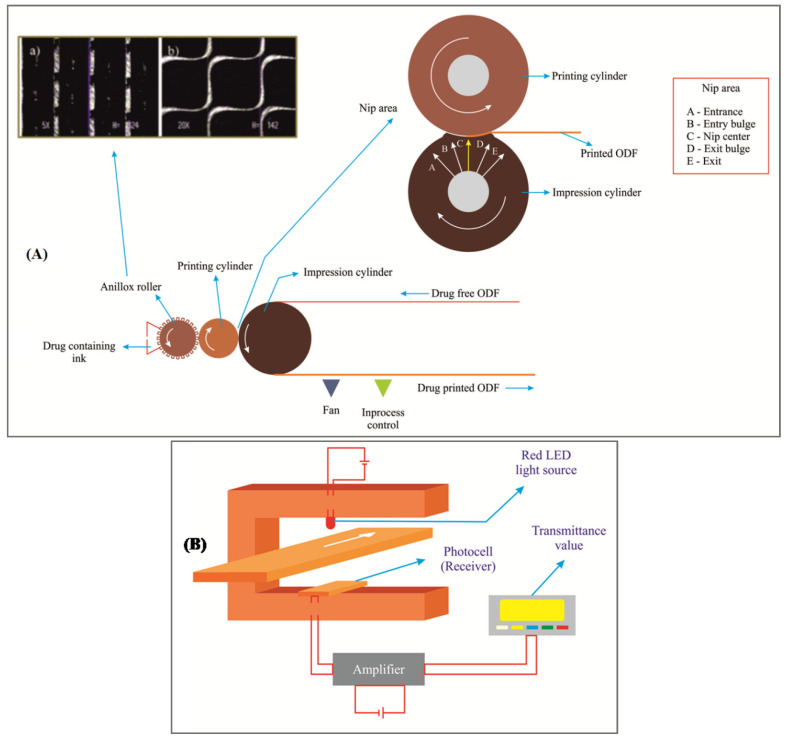
Flexographic printing:(**A**) flexo process showing micrographs of anilox roller and nip area between the printing cylinder and impression cylinder [101]; (**B**) flexo printing with online monitoring—Adapted with permission from reference [101], 2013, Elsevier. All images reprinted with permission from respective publishers.

## 5. Scientific Prospection

A scientific prospection using PubMed database from 2000 to 2020 was carried out using five search queries/taxonomy including (a) orodispersible film or fast dissolving film; (b) orodispersible film and solvent casting; (c) orodispersible film and electrospinning; (d) orodispersible film and extrusion; (e) orodispersible film and printing in ‘Title/Abstract’. The results are shown in Figure 20. There were no hits prior to 2010. A total of 144 publications were found on ‘orodispersible film’. The lowest were recorded in 2010 with 3 publications, while the highest were recorded in 2018 with 31 publications, which is more than double the number that was recorded in 2017 with 15 publications. In 2019, this number plummeted to 19, which later on picked-up nicely in the following year with 30 publications. This gives an impression that there is a slow and steadily growing interest in the domain of ODFs and more publications are expected in the coming years.

The Boolean operator ‘AND’ was used in different combinations of the taxonomy mentioned under (a) to (e). This search was carried out to understand the most preferred fabrication methods employed to prepare ODFs. A total of 41 publications were found, of which 19 were for solvent casting, 12 for printing, 6 and 5 hits were obtained towards extrusion and electrospinning methods of fabricating ODFs. From these hits, we can infer that solvent casting is the most popular and highly used method of fabricating ODFs since 2010.On the other hand, fabrication of ODFs by printing methods has seen the first publication in 2013 with 1 publication. Last year, 4 publications were recorded. Accordingly, it can be inferred that more research is warranted in this domain and it is definitely a potential area for further exploration from a personalization point of view. As regards extrusion method of fabricating ODFs, only 6 publications were found and this is probably because of stability issues due to high drug processing temperatures.

## 6. Conclusions

In general, the last two decades and particularly post-2010 has seen a significant growth in employing different fabrication technologies to prepare ODFs. SCM has been vividly explored for a long time and consequently, more literature in the form of patents and journal publications are available when compared with other fabrication technologies. On the other hand, printing techniques to fabricate ODFs is the upcoming technology and a proper regulatory rigor in this domain would motivate pharmacists to employ this technology in a pharmacy setting to offer personalized medication for patients. Electrospinning stands out as a less explored fabrication technology and, therefore, it requires extensive developmental diligence to help fabricate ODFs of all classes of drugs. Similarly, melt-extrusion is yet another upcoming technology to fabricate ODFs. While the temperature sets a few limitations, recent developments in this domain have aided in overcoming this limitation as well. Combinations of the above known methods are the new trend to fabricate ODFs but are yet to see promising outcomes, and more research is warranted in this domain. Despite the progress seen in the patented technologies to fabricate ODFs, some of the limitations of these technologies are still hindering large-scale production and commercialization. Accordingly, more modifications are envisaged in the existing technologies to produce ODFs of high quality using an effective and efficient process to obtain more output with a minimum number of defects.

## Figures and Tables

**Figure 1 pharmaceutics-14-00820-f001:**
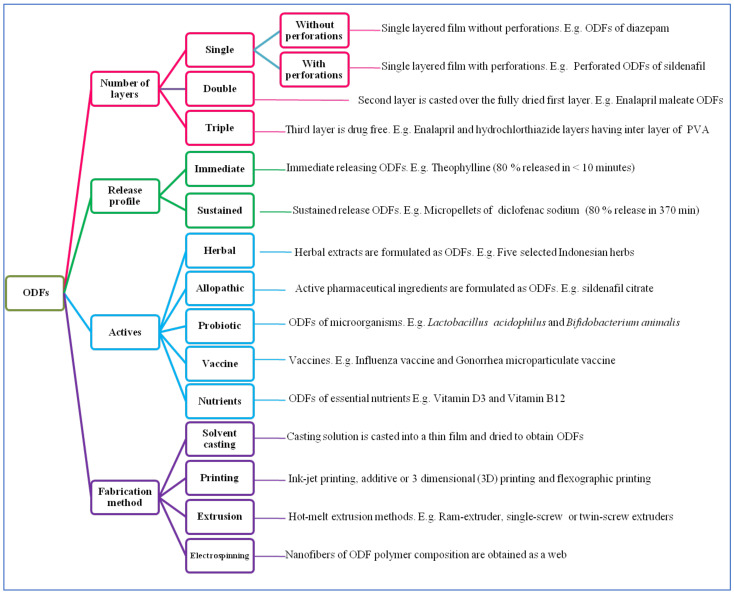
Classification of orodispersible films (ODFs) [5,6,7,8,9,10,11,12,13,14,15,16,17,18,19,20,21,22,23].

**Figure 2 pharmaceutics-14-00820-f002:**
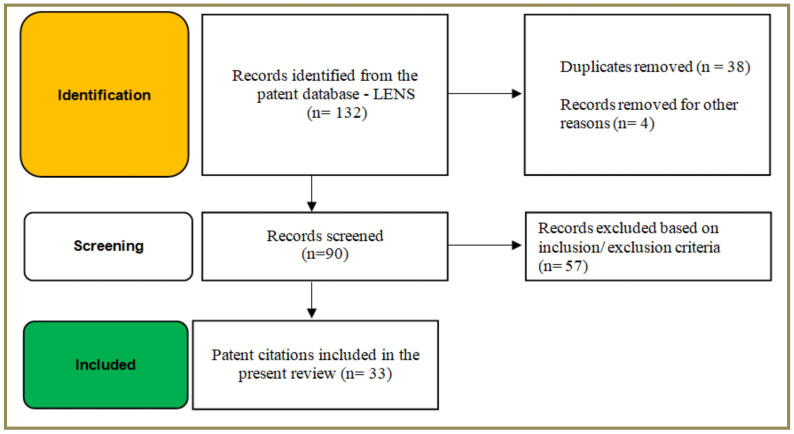
Identification of patent citations using LENS (free online patent database).

**Figure 3 pharmaceutics-14-00820-f003:**
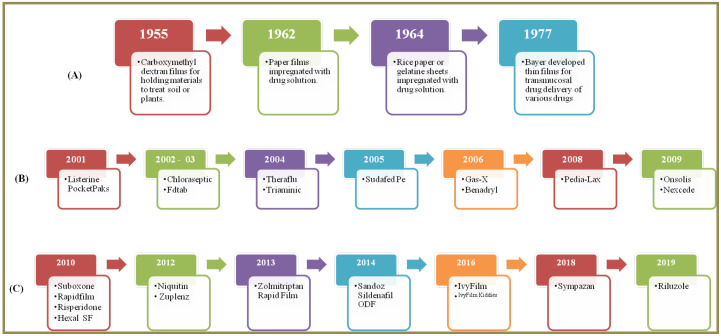
History of Orodispersible films (ODFs) (**A**) prior to 2000 (**B**) over the counter (OTC) products as thin films (2001 to 2009) (**C**) prescription-Rx-products as thin films (2010 to 2020).

**Figure 4 pharmaceutics-14-00820-f004:**
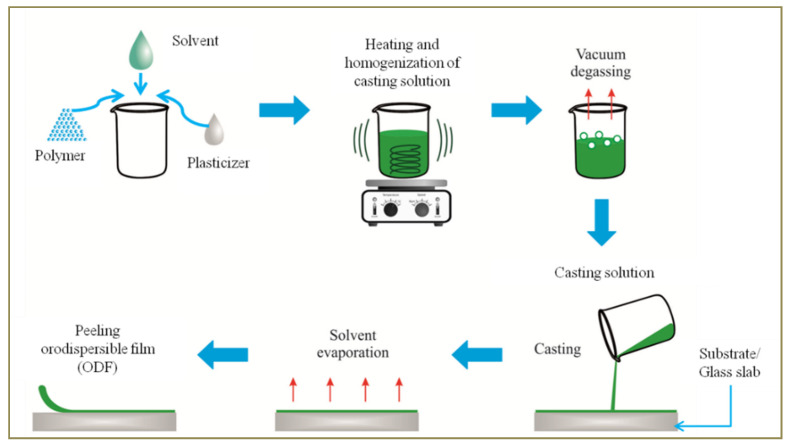
Steps in fabrication of orodispersible films by solvent casting method.

**Figure 5 pharmaceutics-14-00820-f005:**
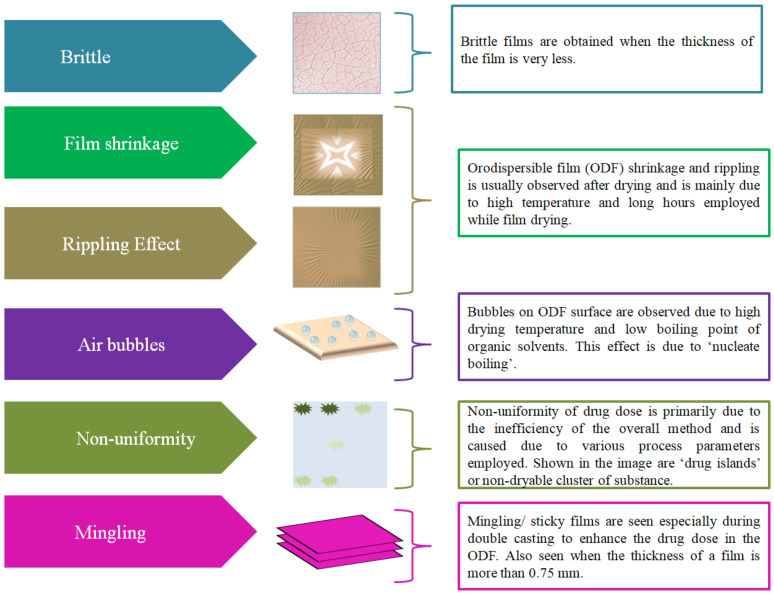
Inaccuracies inorodispersible films fabricated by solvent casting method.

**Figure 6 pharmaceutics-14-00820-f006:**
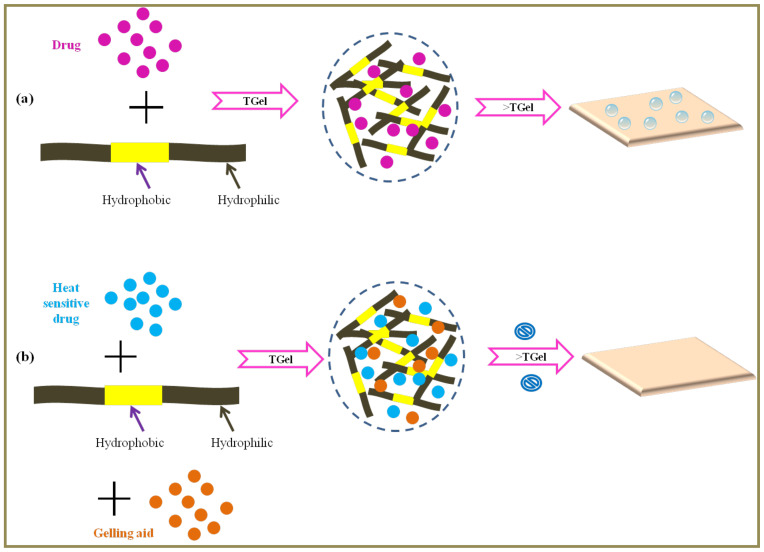
Thermal gelling point (**a**) heating the mixture at higher thermal gelation temperature (TGel) leads to formation of air bubbles in the film (**b**) employing gelling aids reduces the TGel and reduces formation of air bubbles in the film.

**Figure 7 pharmaceutics-14-00820-f007:**
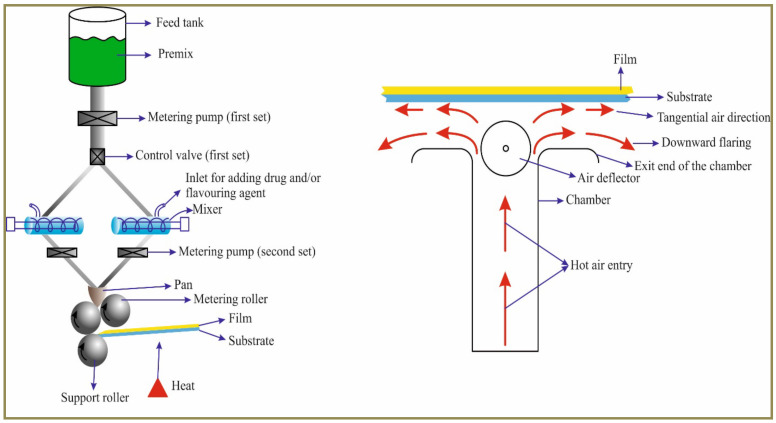
PharmFilm technology to fabricate orodispersible films. Reprinted with permission from [40], 2021, Elsevier.

**Figure 8 pharmaceutics-14-00820-f008:**
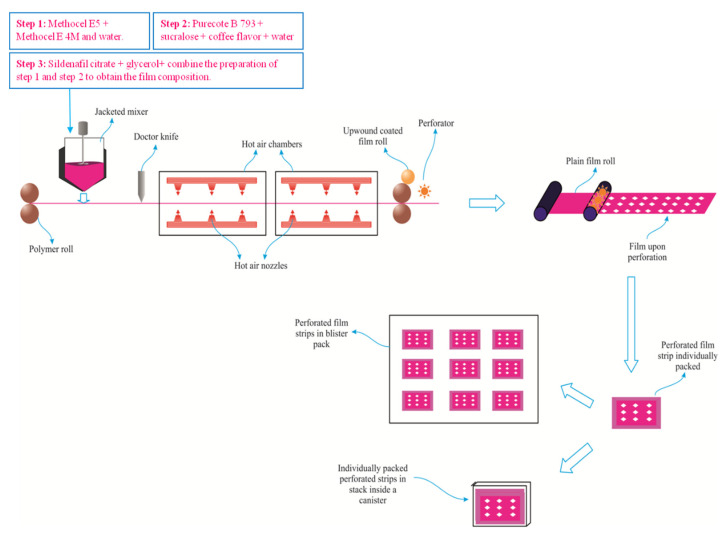
Fabrication of perforated orodispersible films.

**Figure 9 pharmaceutics-14-00820-f009:**
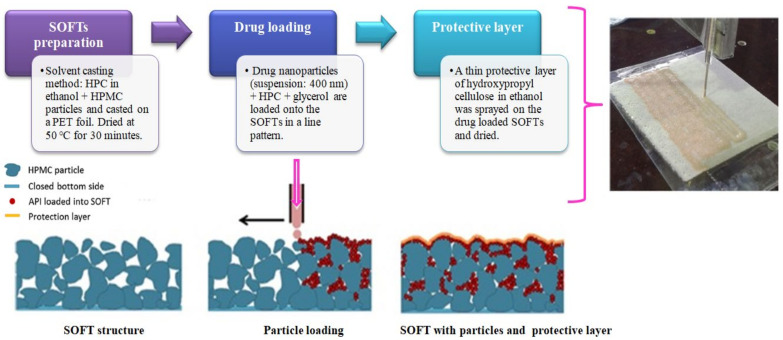
Structuredorodispersible films (SOFTs) and its preparation [66]. Adapted with permission from [66], 2019, Elsevier.

**Figure 10 pharmaceutics-14-00820-f010:**
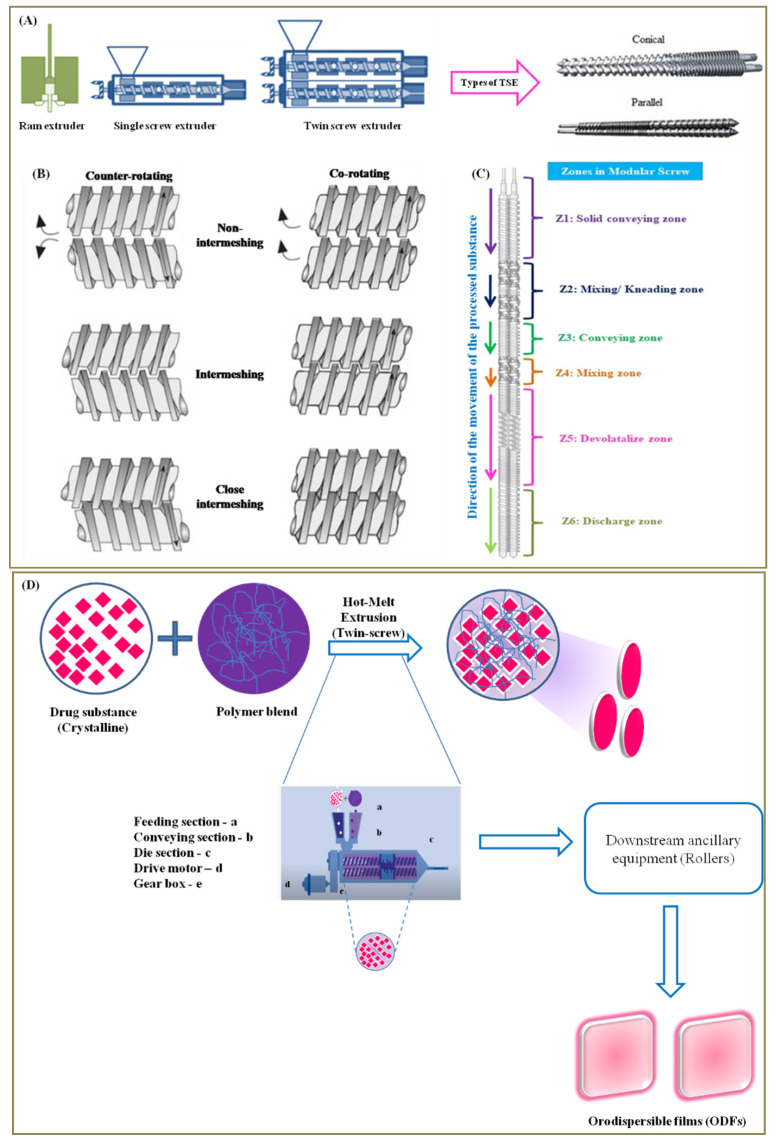
Hot-melt extrusion: (**A**) different extruders and types; (**B**) counter rotating and co-rotating screws; (**C**) different zones on a modular screw; (**D**) process to fabricate orodispersible films (ODFs) by the HME method.

**Figure 11 pharmaceutics-14-00820-f011:**
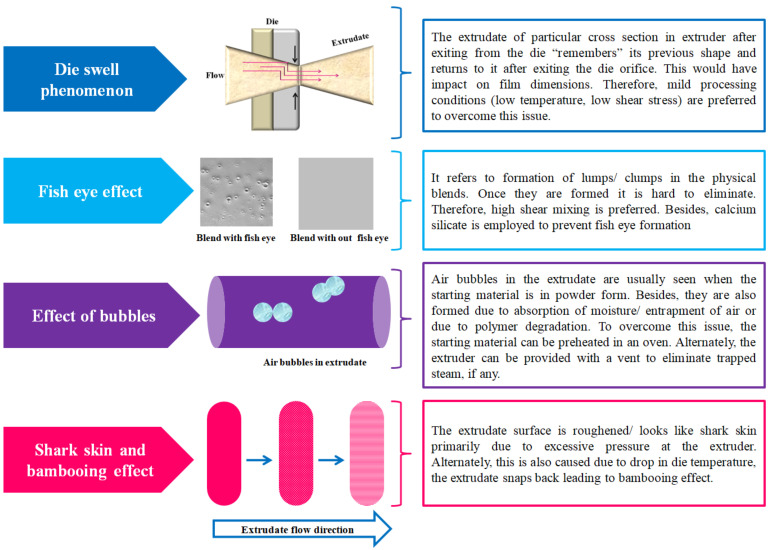
Common limitations of hot-melt extrusion method.

**Figure 12 pharmaceutics-14-00820-f012:**
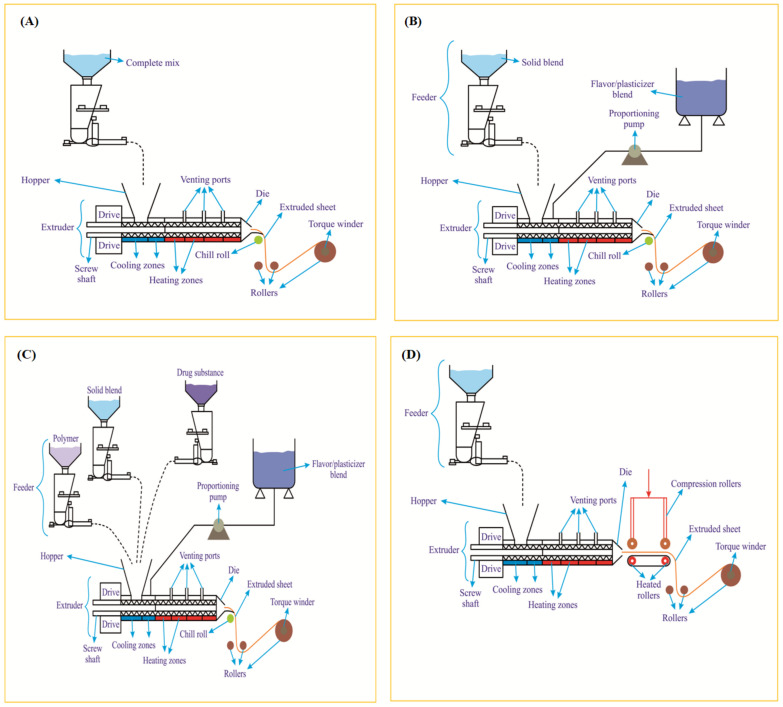
Orodispersible film scale-up by hot-melt extrusion method: (**A**) small scale extrusion—ODF composition can be directly feed into the hopper from the feeder; (**B**) medium scale extrusion—ODF composition of solid blend can be feed from one side and the flavor/plasticizers can be feed from the other side; (**C**) large scale extrusion—depending on the type of ingredients employed in ODF preparation—they can be feed separately from respective feeders to the hopper; (**D**) extrusion process with optional compression rollers and heated rollers to fabricate oral films of nicotine/snuff (tobacco)—Adapted from [85].

**Figure 13 pharmaceutics-14-00820-f013:**
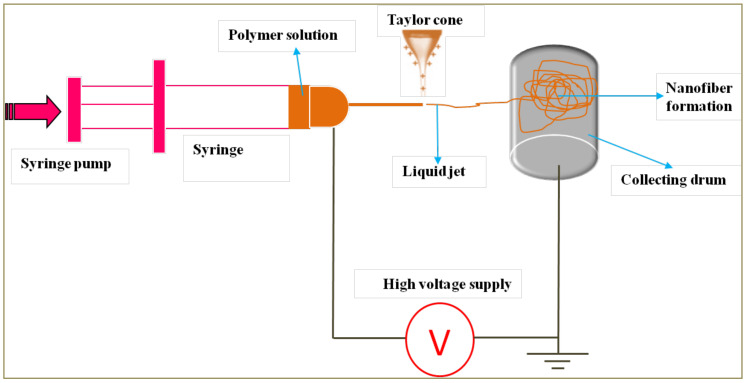
Basic setup of the electrospinning process.

**Figure 14 pharmaceutics-14-00820-f014:**
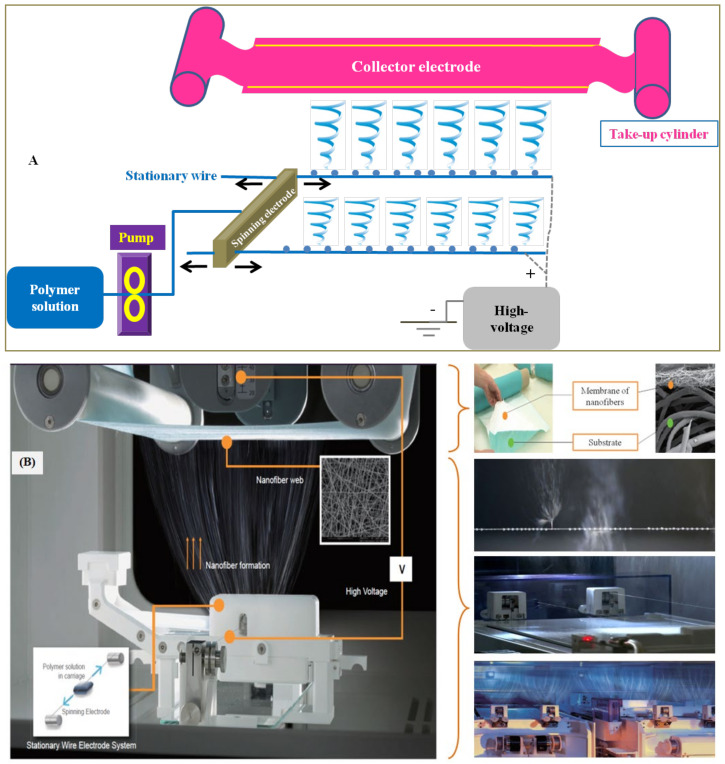
Nanospider technology: (**A**) stationary-wire electrospinning; (**B**) Elmarco’s commercialized Nanospider technology showing nanofiber web on the substrate, stationary wire with polymer solution on the wire and nanofiber formation, stationary wire electrode system and nanofiber formation when fully operational. Image courtesy: Elmarco.

**Figure 15 pharmaceutics-14-00820-f015:**
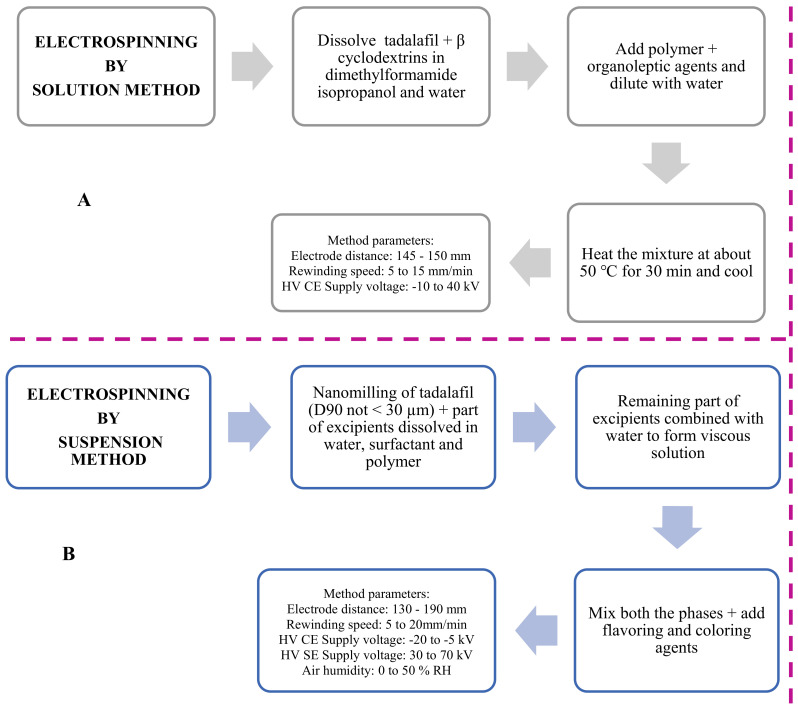
PatentedElectrospinning by (**A**) solution method; (**B**) suspension method.

**Figure 16 pharmaceutics-14-00820-f016:**
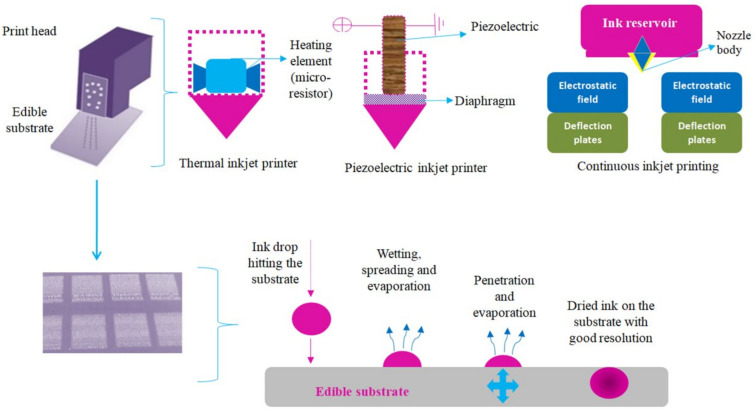
Inkjet printing of orodispersible films showing different print heads, printed ODF and mechanism of ink ejection onto the edible substrate and drying to obtain ODFs.

**Figure 17 pharmaceutics-14-00820-f017:**
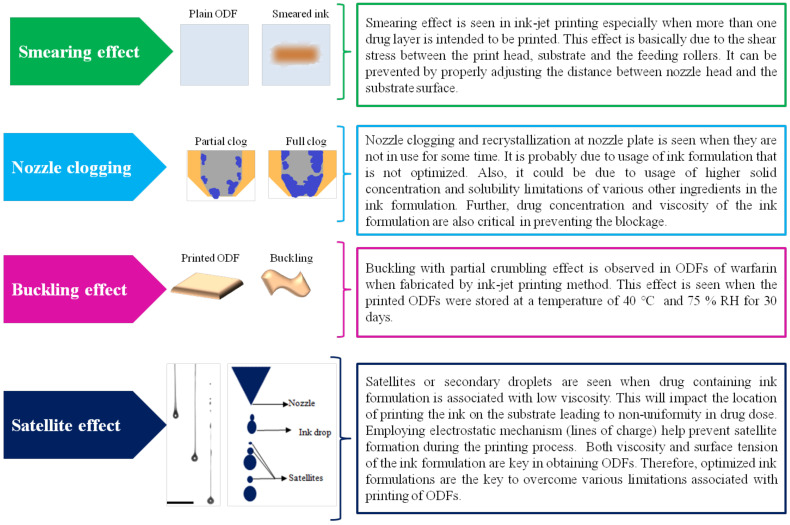
Common limitations of printing methods.

**Figure 18 pharmaceutics-14-00820-f018:**
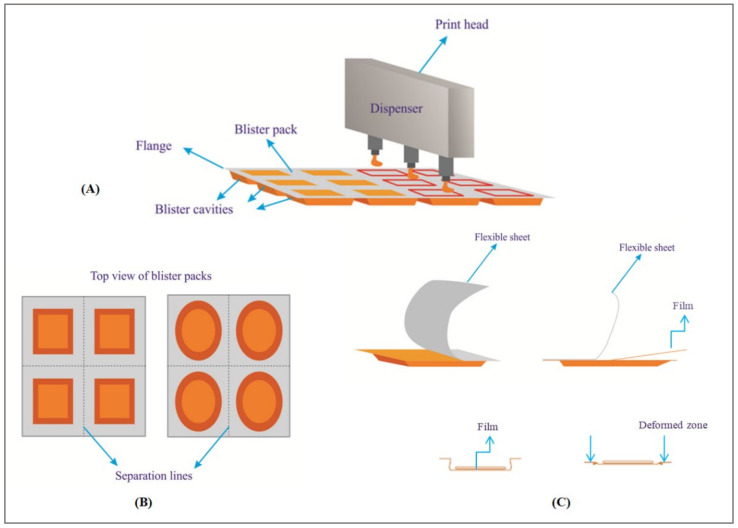
Ink-jet printing: (**A**) printer head dispensing/printing the drug containing ink into the blister cavities; (**B**) top view of the blister packs with inkjet printed ODFs having each dose separated by a separation line; (**C**) individual packaging unit showing a flexible sheet and the packed film—showing zones of deformity during the opening of the film—Adapted from [99].

**Figure 20 pharmaceutics-14-00820-f020:**
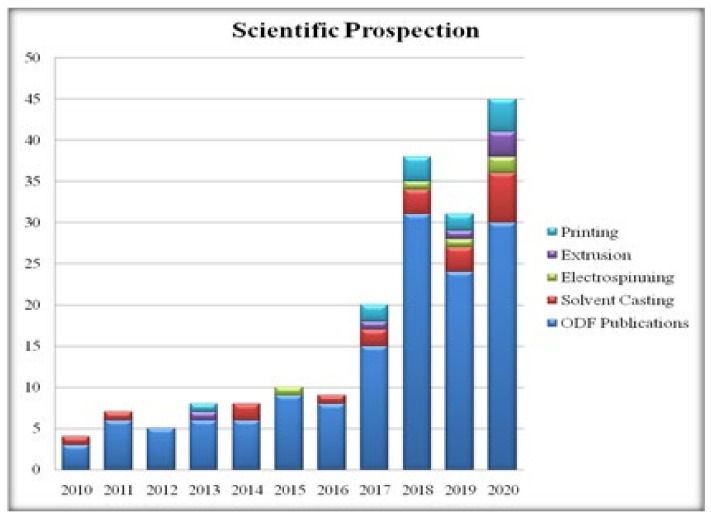
Scientific prospection using PubMed.

**Table 1 pharmaceutics-14-00820-t001:** Taxonomy of patented fabrication technologies.

Level 0		Level I		Level II	No. of Hits Included
ODFFabricationmethods	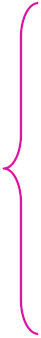	Solvent casting method	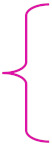	Polymer	**22**
Plasticizer
Solvent
Casting solution
Casting
Rolling
Rollers
Electrospinning method	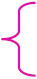	Polymer solution	**3**
Taylor cone
Nanofibers
Collecting drum
Extrusion method	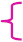	Hot-melt extrusion	**4**
screw, extruder
Printing methods	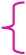	2D printing	**4**
3D printing
Flexographic

**Table 3 pharmaceutics-14-00820-t003:** Composition of a batch for fabricating dextromethorphan hydrobromide orodispersible film by hot-melt extrusion.

Sl. No.	Component	%Weight	HME Process Parameters
1	Dextromethorphan hydrobromide coated granules	30.0	Feeding rate of powder blend: 0.758 kg/h Side stuffer—drug substance flow rate: 0.379 kg/h Side stuffer—feeding screw speed of drug substance: 100 rpm. Peristaltic pump speed: 04 rpm Extruder barrel temperature: 55 °C Die temperature: 65 °C Extruder screw speed: 125 rpm Gear pump speed: 15–22 rpm Film die gap: 0.70 mm Calendar roll gap: <0.1 mm Calendar roll temperature: 35 °C
2	Polyethylene oxide (molecular weight 100,000)	22.8	
3	Mannitol	22.8	
4	Sucralose (micronized)	1.0	
5	Cherry flavor (granules)	13.4	
6	PEG 400	10.0	

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
