# Peer review of "A Comprehensive Review of Patented Technologies to Fabricate Orodispersible Films: Proof of Patent Analysis (2000–2020)"

_pharmaceutics, 2022, doi:10.3390/pharmaceutics14040820_

Round 1

Reviewer 1 Report

I have read the article “A Comprehensive Review of Patented Technologies to Fabricate Orodispersible Films: Proof of Patent Analysis (2000 – 2020) and Hype Cycle” intended for publication in Pharmaceutics carefully, and in my view, it brings an interesting overview about the patent situation in the field of ODF manufacturing and fabrication technologies, supplemented with the hype cycle model.

Specific comments and suggestions:

  1. The manuscript is too extensive and should be shortened.
  2. There are many formal mistakes (e.g., typos, capital letters, structure and numbering of chapters and subchapters, etc.). These mistakes lower the scientific value of the manuscript and should be thoroughly corrected.
  3. In some parts, the English language does not sounds “scientific.”
  4. The arrangement of the industrial line used for ODFs manufacturing by solvent casting method should be described and presented. Discussed is mainly the lab arrangement for small batches and R&D.
  5. What is the principle (reason) of film shrinkage (p.7, L238-240) since it is a very common problem. Discuss please this in more detail.
  6. L237 “thickness of the film is very LOW.” Similar mistakes (inappropriately used words and terms) could be found in many places of the manuscript (e.g., accommodate higher drug dose, malitol, Flavorant, Enalapril, low disintegration time, etc.).
  7. Please discuss more in detail the creation of film perforations mentioned in L444.
  8. Sections 4.1.6. and 4.1.7., there is nothing about the technology, just casting dispersion.
  9. Section 4.1.8. and 4.1.11. should be discussed more in detail.
  10. Same units should be used throughout the whole manuscript.
  11. Section 4.4. there is a confusing division of printing technologies.
  12. 3D printing for ODF manufacturing should be as the modern trend discussed in more detail.

The presented manuscript could be a valuable source of information for scientists and manufacturers interested in ODF preparation, but its quality has to be improved. I do recommend major revision based on the above comments.

Author Response

Please find our answers to the reviewer queries in the attached file.

Reviewer 2 Report

Dear Authors, I have carefully read your manuscript entitled “A Comprehensive Review of Patented Technologies to Fabricate Orodispersible Films: Proof of Patent Analysis (2000 – 2020) and Hype Cycle.

It is quite interesting work with a lot of images, diagrams and charts. In my opinion, it has two parts of uneven scientific value:

  • Patent analysis
  • Hype cycle

In the first one, you have analyzed recent patents on the fabrication of the orodispersible films in excellent way. This part is scientifically sound and may be published with some minor corrections and additions.

I have the following questions and remarks to this part:

  1. In figure 1 you have classified ODFs printed by the Jamroz et al [Ref. No 9] as a multiple layer films. However, these films has only single layer. Indeed, they were printed by the fused deposition modeling (FDM) technology which comprises printing 3D objects layer-by-layer. However, in order to achieve the shortest disintegration time, the printed films were built only by one layer, the thinnest possible to print with their equipment, i.e. 150 µm. Therefore, these films cannot be classified as a multilayer ones.
  2. Page 5, line 197 - In the description of the casting method, you have stated that “the casting solution is spread over the clean and smooth surface (glass slab)”. The same information is placed on the figure 4 (page 6). However, in the description of the patented technologies, there is usually not the glass slab, but the ‘substrate’ or the ‘plain film roll’. I think that you should also put information on these two materials along with the ‘glass slab’’ in the general description of the method.
  3. Page 7, lines 257-270 – description of UNM Rainforest Technology does not include any information on the film properties, e.g. disintegration time, mechanical characteristics, etc. It would be very valuable to add such data, if it is available.
  4. Page 11, lines 422-424 – the statement “Hexal’s patented technologies provided evidence to prove the fact that ODFs form fewer impurities during storage when compared with tablets that tend to form more impurities over a period of time” seems to be too general.

It is hard to believe, that the amount of impurities in the tablets prepared by direct tableting or the tableting after dry granulation (compaction) may be higher than in the casted ODFs. You based this statement on the patent work, where Hexal compared ODFs with coated tablets. Therefore, the amount of impurities presented in the tablets might be higher than in the ODFs, because of the coating process (heat and water) or because there was wet granulation prior to the tableting, which may also lead to the formation of the impurities.

  1. Page 11, lines 432-450 – I didn’t find information in your description, about the reason for film perforation? Is it made to shorten disintegration time? You could add some information on this.
  2. Page 12, lines 479 – 502 – the same as in point 3. If there is any available data on the film properties prepared with these technologies, it would be nice to add it to the text.
  3. Page 13, lines 508-513 – magnesium oxide and sodium hydroxide were used for the taste-masking. Could you provide more details on the mechanism of taste masking with these two components? Is it just because the lower solubility of the sildenafil in the higher pH?
  4. Page 13, lines 520-521 – “Increase in pH was helping to improve disintegration time” – could you comment why? Was there the effect of pH on the solubility of the pullulan?
  5. Table 2, patent no. 2 – there seems to be a wrong reference number [40]. In the reference section, the number [40] refers to the scientific article of Gupta et al., not the patent number.
  6. Page 4 (18) in section 4.2. (There is some mess with number of the pages, some of them are doubled, and not in order. There is also no number of the lines on this page) – in the bottom of the page you enlisted common limitations of the hot-melt extrusion process. However, there is no information on the high temperature of the process, which may lead to the degradation of the API and is considered one of the most important limitations of this method. Another one is the lack of the possibility to use any volatile ingredients, e.g. oils as flavorants.
  7. Figure 10 C – I suggest adding the arrow which shows the direction of the movement of the processed substances between the screws. It may be not obvious that Z1 is the first zone and Z6 the last one.
  8. Figure 10D – in the middle of the figure there is a scheme with numbers from 100 do 140. Are they just numbers referring to the captions in the Legend? Why aren’t they simpler, like a), b), c), d), etc…..

In such form, they may suggest the values of the temperature in the hot-melt extrusion process.

  1. Page 3 (21) – the hot-melt extrusion process patented by Joseph Fuisz refers neither to the medicinal product, nor the orodispersible films. It describes the tobacco sustained release thin films for recreational use. Why was it placed in the text?
  2. Temperature units should be the same in the entire text. In one place there are Fahrenheits, in other Celsius.
  3. Page 5 (23), line 168-169 – You have stated that “the critical requirement to employ the electrospinning method for ODFs is the preparation of final formulation in the form of a solution.” However, in the next paragraph you describe the electrospinning of the suspension. So, is the solution necessary for this process, or not?
  4. Page 7 (25), line 239 – You have stated that the most modern method of fabricating ODFs is by printing technologies. However, in the figure 24 and 25, the first patent on ink-jet printing is dated 2008 while electrospinning on the 2011. So it seems that the electrospinning is the most modern method, not the printing.
  5. Page 10 (28), line 335 – please provide the full name of the EPDM. There is only abbreviation.
  6. Page 10 (28), line 351 – you just mentioned the 3D printing method used for the preparation of ODFs. What kind of technology was that? (fused deposition modeling, laser sintering, etc.)
  7. Page 11 (29), line 388 – ‘fabrication of ODFs by printing methods has seen the first publication in 2017’. This is not true. There were earlier publications on the application of printing to the preparation of ODFs, e.g. Eva Maria Janßen “Drug-printing by flexographic printing technology—A new manufacturing process for orodispersible films” International Journal of Pharmaceutics, Volume 441, Issues 1–2, 2013, p. 818-825, https://doi.org/10.1016/j.ijpharm.2012.12.023.
  8. Page 14 (32), figure 23 – there is a statement “Yet to be included in the official compendia of any country” – ODFs were defined for the first time in the European Pharmacopoeia 7.4 in 2012.
  9. Figures 24 and 25 present almost the same data. One of them should be removed, in order to not duplicate the same information.
  10. There are also some misspelled words and grammar errors through the entire text:
    - page 7, line 237 – expression “very less” is not correct

- page 9, line 360 – is “addition f” instead of “addition of”

- page 10, line 365 – there is “neuroleptics drugs”, should be “neuroleptic drugs”

- page 10, line 398 – there is “Flavorant”, should be “flavorant”

- page 10, line 407 – there is “inorder” instead of “in order”

- page 3 (21), line 78 – there is “0.05 to millimetres”, should be “0.05 millimetres”

- page 7 (25), figure 15 – there is “D90 not > 30 µm”. I suggest “D90 < 30 µm”

- page 18 (36), line 662 – there is “platreau”, should be “plateau”

  1. The references to the patents should be changed. There is no information on the authors of inventions.

The second part consists of the application of the “Gartner Hype Cycle” to the description of the ODFs evolution. This approach seems to be not very scientific in nature, and is just the subjective opinion of the life cycle of ODFs technology.

In my opinion, there is no data presented in the manuscript that objectively supports the particular stages of the cycle.

For example, you place at the peak of the cycle, the year 2010, which suggest that after that the visibility of the ODFs significantly decreased (“sliding into trough”). However, when you consider the number of scientific publications or the patents, there is no such decrease.

Actually, the publication number just started to increase in 2010.

The most objective data would be the data on the annual salary or revenue of ODFs. You have only presented selected information on the economic factors and all of them are growing numbers. There is no “sliding into trough” in them.

Therefore, I think that this part of your manuscript may be good for a popular science journal or for business opinion on the ODFs technology, but it is not scientifically sound enough to be published in prominent scientific journal, and to be treated as a real evidence of the lifecycle of this technology.

I suggest removing of the entire part describing Gartner hype cycle. Some information can be moved to the Introduction section or to the chapter 3.0 History.

Author Response

(The authors gave the same response as above.)

Round 2

Reviewer 1 Report

Dear authors,

the manuscript was improved significantly, and I recommend its publication in the presented form.

Reviewer 2 Report

Dear Authors,

Thank you for your response letter to the review. I have read the corrected manuscript. I think that it is much improved and may be published.
I have still one remark regarding the patent references. According to the MDPI citation guideline:

MDPI’s style for citations and references lists are widely based on the style used by the American Chemical Society. Please refer to the ACS Style Guide [1].

Here is the proper citation example according to the ACS:

Patents
Gordon, A. Z.; Rossof, A. H. Methods for Treating Leukopenia with Tertiary Amines or Quarternary Ammonium Salts. U.S. Patent 5,466,509, 1987.

Fritzberg, A. R. Metal Radionuclide Labeled Proteins for Diagnosis and Therapy. Eur. Pat. Appl. 188256, Jan 13, 1986.

I think that omitting inventors names in the references list may neglect their contribution.

This is obviously editor's decision whether it is acceptable, but in my opinion the inventors names should be included in the references.